# Unveiling the Interplay—Vitamin D and ACE-2 Molecular Interactions in Mitigating Complications and Deaths from SARS-CoV-2

**DOI:** 10.3390/biology13100831

**Published:** 2024-10-16

**Authors:** Sunil J. Wimalawansa

**Affiliations:** CardioMetabolic and Endocrine Institute, North Brunswick, NJ 08902, USA; suniljw@hotmail.com

**Keywords:** 25(OH)D, 1,25(OH)_2_D, calcitriol, endocrine, mechanisms, morbidity and mortality, pandemic, public health, renin-angiotensin axis

## Abstract

The SARS-CoV-2 virus that caused COVID-19 devastated families, social structures, and economies worldwide. This pandemic has overwhelmed healthcare systems, increased deaths and disabilities, and triggered a global socio-economic crisis. Although the COVID-19 vaccines were developed rapidly, their effectiveness significantly decreased by the end of 2021 due to mutated viruses evading the immune system. As a result, despite high vaccination rates in industrialized countries, significant outbreaks occurred due to immune evasion associated with viral mutations. Over 300 clinical studies have shown that vitamin D (and ivermectin) are widely available and economical agents that promote immune system function. Proper doses of vitamin D effectively prevent and treat SARS-CoV-2, reducing complications, hospitalizations, and deaths by approximately 50%. Those with vitamin D deficiency fare the worse. SARS-CoV-2 activates the renin-angiotensin system by increasing renin expression, leading to elevated levels of the inflammatogenic and vasoconstrictor peptide angiotensin-II. SARS-CoV-2 viruses cause widespread inflammation, blood clots, and lung damage through multiple mechanisms, leading to impaired tissue oxygenation and death. In addition to enhancing the immune system, vitamin D increases ACE-2 enzyme levels, which breaks down angiotensin-II and reduces SARS-CoV-2-induced inflammation. It also lowers blood pressure and mitigates abnormal clotting. While the virus enters human cells through ACE-2 receptors, excess ACE-2 spills into the bloodstream and neutralizes viruses. This manuscript discusses how vitamin D mitigates the harmful effects of COVID-19.

## 1. Introduction

The COVID-19 pandemic, triggered by the severe acute respiratory syndrome coronavirus-2 (SARS-CoV-2), caused havoc worldwide, leading to socio-economic crises and widespread negative repercussions on the global populace [1,2]. Vitamin D is a secosteroid molecule that undergoes two steps of hydroxylation to generate its most active form, calcitriol [1,25-dihydroxyvitamin D; 1,25(OH)_2_D] [3]. Beyond its pivotal role in calcium regulation, vitamin D engages in various biological functions affecting all tissues, especially the modulation of innate and adaptive immunity [4]. Upon binding to the vitamin (calcitriol) D receptor (VDR, CTR), it forms a complex with cofactors, translocates into the nucleus, and attaches to relevant portions of DNA [5]. While vitamin D and its receptor polymorphisms affect disease vulnerability and responses [6,7], they regulate over 1700 human genes, up- or down-regulating target genes [8].

The human immune system is operated by a complex network of mechanisms that respond to signals generated by membrane-bound signaling molecules, including toll-like receptors (TLR) [9,10,11]. In humans, many overlapping mechanisms effectively regulate the innate and adaptive immune systems [12]. Over 75% of the immune system functions rely on having sufficient calcitriol synthesized within immune cells. Consequently, vitamin D modulates immune cell functions and helps maintain a robust immune system [5,13]. Moreover, immune cells themselves express both vitamin D/calcitriol receptor (VDR/CTR) [14] and the 1α-hydroxylase enzyme (from the *CYP27B1* gene) responsible for converting 25(OH)D into 1,25(OH)_2_D intracellularly [11,15,16]. Beyond its classical genomic functions, vitamin D exerts membrane-based and non-genomic actions, swiftly modulating various physiological pathways. These rapid actions and their autocrine and paracrine functions are particularly prominent within immune cells [17].

Vitamin D modulates the immune response, potentially reducing inflammation, and is associated with cytokine storm in severe COVID-19 [18]. Vitamin D up-regulates angiotensin-converting enzyme-2 (ACE-2) expression and generates angiotensin_(1–7)_ (Ang_(1–7)_), a potent vasodilatory peptide that counteracts angiotensin-II (Ang-II). Ang_(1–7)_ protects the cardiovascular system and mitigates lung injury caused by coronaviruses. Understanding the molecular interactions of these actions will provide insights into novel therapeutic strategies and public health interventions to reduce the burden of COVID-19. This review explores the interactions of vitamin D and ACE-2 on the backdrop of COVID-19 that affect clinical outcomes by critically evaluating ongoing research and recently published studies in this critical area.

This review article highlights the complex relationship between vitamin D and the ACE-2 receptor, focusing on their roles in mitigating the severity and mortality of COVID-19. We hypothesized that ACE-2 has a vital beneficial role in mitigating complications and infections of COVID-19 despite its role as the primary entry site for coronaviruses into epithelial cells. This study also investigated the vitamin D-dependent mechanisms of ACE-2 in inhibiting SARS-CoV-2 replication and subduing inflammation and oxidative stress reduction. This review highlights scientific data on the interactions of vitamin D on the expression and function of ACE-2 related to SARS-CoV-2 entering human cells and mitigating its harmful effects [19].

### 1.1. Benefits of Vitamin D

Research has shown that vitamin D is essential for more than just bone health. It plays a role in (A) the musculoskeletal system—it prevents rickets in children and osteomalacia in adults) [20,21,22], and (B) immune functions—vitamin D is crucial for maintaining a robust immune system [23,24] and reducing the risk of infections and autoimmune diseases [4,25], modulates metabolism [26] and energy generation [27,28], prevents acute infections [4,22], and minimizes chronic diseases [29]. Adequate vitamin D levels are linked to a lower risk of several chronic conditions, such as cardiovascular disease [30,31] and certain cancers [32,33], but others disagree [34,35]. In addition, it minimizes pregnancy-related complications and disorders [36,37,38,39] and mental health conditions [40,41], potentially including bipolar disorder [42] and the prevention of depression [43,44].

Meta-analyses and other clinical studies have consistently shown that sufficient vitamin D supplementation protects against acute respiratory tract infections, particularly in individuals with significant vitamin D deficiencies [45,46]. Further clinical research, including randomized controlled trials (RCTs), has revealed that vitamin D supplementation plays a crucial role in achieving positive health outcomes in cases of infectious diseases [13,47]. There are more than 310 peer-reviewed publications on the beneficial effects of prophylactic vitamin D in preventing SARS-CoV-2 [48] and as an adjunct in treatment [43,49,50]. Figure 1 illustrates the broader benefits of vitamin D in human health related to the immune system.

Vitamin D is a threshold nutrient. Consequently, there will be little additional benefit from over-supplementation or intake by those with sufficient vitamin D [5]. However, supplementing deficient individuals provides a significant benefit [52]. In parallel, over-exposure to sunlight does not lead to excess vitamin D entering the blood [53]. Notably, for those infected with SARS-CoV-2 (and other viruses), vitamin D and its metabolites are consumed rapidly during infections and conditions associated with immune system activation [51,54]. Therefore, unless supplemented, even those with reasonable serum 25(OH)D levels (e.g., on hospitalization) are likely to develop a deficiency from an acute severe infection like SARS-CoV-2 [5]. 

Notably, over half of the world’s population is vitamin D deficient at any given time [5]. Therefore, with little or no adverse effects, it is beneficial to consume vitamin D supplements or be exposed to safe levels of daily sunlight [55,56,57]. However, even in the sunniest regions in the world, the majority spend time indoors. Thus, most people need supplemental vitamin D to prevent deficiency to stay healthy.

### 1.2. The Entry of SARS-CoV-2 into Human Cells

SARS-CoV-2, the virus responsible for COVID-19, enters human cells primarily by its spike (S) protein interaction with the angiotensin-converting enzyme 2 (ACE-2) receptor on the surface of human cells [57]. The spike protein has two subunits: S1, which contains the receptor-binding domain (RBD) that specifically binds to ACE-2, and S2, which facilitates membrane fusion [58,59]. Upon binding to ACE-2, the spike protein undergoes a conformational change, enabling the virus to attach more firmly to the host cell. Transmembrane protease serine 2 (TMPRSS2) in the host cell cleaves the S-protein at specific sites, triggering the fusion of the viral and cellular membranes and allowing the viral RNA to enter the cytoplasm [60].

Once inside the host cell, the host’s ribosomes translate the viral RNA into viral proteins. The viral RNA genome serves as a template for replication and transcription to produce more viral RNA genomes and sub-genomic RNAs [61], which encode structural and accessory proteins [60,62]. These newly synthesized viral components are assembled into new virions in the host cell’s endoplasmic reticulum-Golgi intermediate compartment [63]. The mature virions are transported to the cell surface in vesicles and then released into the extra-cellular space via exocytosis, including the tissue fluid and circulation. These virions are primed to infect new local and distant host cells. This viral entry, replication, and release cycle leads to the spread of SARS-CoV-2 within the host and contributes to the pathology of COVID-19 [64].

Meanwhile, the expression of ACE-2 is regulated by vitamin D metabolites [65]. Vitamin D sufficiency increases the expression of ACE-2 (including the soluble ACE-2 in the circulation) and dampens the renin-angiotensin system/axis (RAS) [59]. Despite cell membrane-bound ACE-2 receptors serving as the primary entry point for the SARS-CoV-2 virus, numerous benefits of vitamin D interaction have been reported, initiating ACE-2-related anti-viral effects against SARS-CoV-2 [19]. This review investigates the impact of vitamin D on the immune system, focusing on its effects on the renin-angiotensin system (RAS) and ACE-2 [66].

### 1.3. Functions of the Renin-Angiotensin System (RAS)

The renin-angiotensin-aldosterone axis (RAS) is a crucial endocrine axis that controls critical physiological parameters, such as blood pressure homeostasis, immune functions, and metabolism [66]. It is also known as the renin-angiotensin-aldosterone system (RAAS). The RAS is an organized complex hormonal cascade that plays a vital role in controlling critical functions in many organs. Traditionally, RAS plays a crucial role in the cardiovascular system, metabolism, cell growth, and homeostasis.

However, over the past 15 years, additional functions of the RAS, such as ACE-1 and ACE-2) were identified. These include inflammatory processes that lead to lung and other epithelial cell injury, even causing acute respiratory distress syndrome (ARDS), sepsis, cardiac hypertrophy, pulmonary hypertension, acute pancreatitis, and glomerulonephritis [67]. The RAS is crucial in regulating several physiological parameters related to cardiovascular and renal function [66]. Details of these are illustrated in Table 1.

The primary active peptide of the RAS is Ang-II. However, there are other minor components, like Ang-III and Ang-IV, as well as a counter-regulatory peptide, Ang_(1–7)_ [66]. From a physiological point of view, the most important and widely studied two peptides are Ang-II and Ang_(1–7)_ [68], which are involved in human health maintenance and disease statuses. Investigations are ongoing to understand its role in inflammation [18]. The RAAS is integral to the homeostatic regulation of cardiovascular, renal, and fluid balances (homeostasis) and is essential in physiological stability [67]. A sustained imbalance of the RAS can lead to a lowering of ACE-2 (with loss of its protective effects) [59], causing acute lung injury, like acute respiratory distress syndrome (ARDS), with high mortality.

### 1.4. Vitamin D on the Renin-Angiotensin-Aldosterone Hormonal (RAS) Axis

Activating the RAS, including coronaviruses, leads to the excess production of the enzyme renin, which cleaves angiotensinogen into Ang-I. ACE -1 then cleaves Ang-1 to form Ang-II [69]. This over-activation of the RAS leads to the excess production of Ang-II, which enhances the production of inflammatory cytokines [70], worsening the morbidity and mortality associated with infections/sepsis and leading to a cytokine storm that causes lung damage [71]. Excessive production of inflammatory cytokines leads to uncontrolled inflammation and is linked to immune dysfunction, as seen with excess Ang-II, which causes enhanced inflammation and oxidative stress [70]. In contrast, sufficient vitamin D suppresses the expression of renin, thus damping the RAS while increasing ACE-2 and reducing Ang-II. Consequently, it reduces inflammation and oxidative stress, improving clinical outcomes from infection and septic consequences [51,72,73].

As aforementioned, calcitriol is a crucial regulator of the RAS axis and suppresses the expression of the renin gene—a rate-limiting step of the synthesis of Ang-II via cyclic AMP (cAMP)-dependent PKA signaling [74]. Calcitriol independently increases the expression of ACE-2, reducing Ang-II and increasing the generation of angiotensin_(1–7)_, suppressing the formation of the CRE-CREB-CBP complex [70], and keeping RAS activity under control. Meanwhile, certain viral infections like SARS-CoV-2 stimulate RAS activity, aggravating the situation in those with vitamin D deficiency and making them more vulnerable to developing cytokine storms [19,70].

### 1.5. Functions of the ACE-2/Ang_(1–7)_/MasR Axis (Counter-Regulatory Pathway)

Vitamin D inhibits renin, induces ACE-2/Ang(1–7)/MasR axis activity, and modulates the ACE/Ang II/AT1R axis. Through these mechanisms, vitamin D increases the expression of ACE-2, MasR, and Ang(1–7) and elicits a protective role against acute lung injury/ARDS and abnormal clotting activities [75]. Vitamin D can stimulate these beneficial targets like ACE-2 [65,76]. The RAS is regulated mainly by vitamin D (calcitriol) suppressing renin and increasing the expression of ACE-2 [65]—the latter also acts as a primary receptor for SARS-CoV-2 entry into the cells [76].

In health, there is a balance between the RAS, the ACE-Ang–II-AT1R regulatory axis, and the counter-regulatory axis (a pathway) of ACE-2-Ang–1–7-MasR. They are crucial in maintaining human cardiovascular, inflammatory, and immune homeostasis [66]. The wide distribution of ACE-2 in the heart, kidneys, lungs, colon, testis, etc., reflects this. The activation of ACE-2 antagonizes the over-active RAS system, protecting against organ damage and mitigating hypertension and cardiovascular diseases [65]. Once infected, SARS-CoV-2 causes a major imbalance in the RAS, including the down-regulation of ACE-2 receptors when they invade human epithelial cells [75,77].

### 1.6. Molecular Aspects of the RAS and ACE

Angiotensin-converting enzyme-2 (ACE-2) is a peptidase expressed on epithelial cell membranes and plays a crucial role in catabolizing Ang-II, thereby regulating the RAS [78]. Furthermore, in humans, ACE-2 polymorphisms can influence susceptibility to diseases such as hypertension and coronaviruses, including SARS-CoV-2 [78,79]. ACE-2 counterbalances the enzymatic actions of ACE-1 and Ang-II synthesis by converting Ang-II into the peptide Ang_(1–7)_, reducing Ang-II molecular concentration. Furthermore, Ang_(1–7)_ acts via the G-protein-coupled Mas receptor (MasR) to induce vasodilation [80] and attenuate the expression of pro-inflammatory cytokines [68], like TNF-α and IL-6 in LPS-induced macrophages, thereby promoting anti-inflammatory effects [81]. Despite these benefits, functional membrane ACE-2 receptors have been identified as the primary entry site of coronavirus into human cells [82,83].

In addition, Ang II interacts with the adrenogenic axis—endothelin and neuro-adrenergic systems—enhancing the local expression of noradrenaline and endothelin. This pathway leads to trophic as well as adverse effects on the cardiac myocyte [84,85]. Ang II also stimulates aldosterone secretion from the adrenal glands, contributing to salt balances and cardiac and vascular remodeling [84]; these compensatory mechanisms readjust vascular homeostasis. However, continuously raised Ang-II could lead to a loss of compensatory capacity and negatively affect the vascular system. In these situations, ACE inhibitors (ACEi) and higher doses of angiotensin receptor blockers (ARBs) are vital in stabilizing the condition and preventing cardiac failure [85,86].

Additionally, Ang_(1–7)_ has direct anti-inflammatory effects on microglia [87] and contributes to reducing inflammation in adipose tissue [88], as demonstrated in arthritis models [89]. Ang_(1–7)_ also elicits anti-thrombotic actions through the Mas-receptor-mediated release of nitric oxide (NO) from platelets [90]. Therefore, Ang_(1–7)_ also effectively counters and neutralizes the detrimental effects of Ang-II [91]. Ang_(1–7)_ also provides benefits to several major organs. Ang_(1–7)_ attenuates myocyte hypertrophy and cardiac interstitial fibrosis in cardiac tissue [92,93], leading to the higher expression of the ACE-2 gene [85]. In individuals with type-2 diabetes mellitus, Ang_(1–7)_ improves insulin sensitivity, reverses hyperglycemia, and reduces diabetic nephropathy [94].

In the kidney, Ang_(1–7)_ facilitates vasodilation, enhancing renal blood flow and mitigating renal hypertension, preventing further production of the vasoconstrictive Ang-II [95]. Additionally, Ang_(1–7)_ increases the glomerular filtration rate (GFR) and water and electrolyte molecular excretion in a dose-dependent manner [68,96]. Following closed traumatic brain injury, both Ang_(1–7)_ and ACE-2 have improved cognitive and neurological functions [97]. Furthermore, they exhibit protective effects in cerebral ischemia [98] and hemorrhagic stroke in animal models [99,100,101,102].

Moreover, the modulation of the ACE-2/MAS pathway holds promise in preventing pulmonary injuries and represents a potential target for drug development to reduce viral entry [103]. ACE-2 exists in membrane-bound and soluble forms [59]. It is a double-edged sword, as SARS-CoV-2 utilizes membrane-bound ACE-2 to invade epithelial cells [67]. Meanwhile, soluble ACE-2 in extra-cellular fluid, particularly in circulation, binds to coronaviruses like SARS-CoV-2, facilitating their neutralization [104].

## 2. Regulation of the Immune System

Calcitriol, 1,25-dihydroxycholecalciferol, the most active vitamin D metabolite, is crucial for immune cell functions. It is a potent immune modulator. It has been estimated that two-thirds of immune cells’ physiological activation and functions rely on generating sufficient calcitriol within them [13,105]. Because circulatory concentrations are too low, the capacity of calcitriol to diffuse into immune cells is minimal [106]. In contrast, vitamin D and calcifediol 25(OH)D (in nmol) circulate approximately 900 times higher than the hormonal form of calcitriol (pmol). This allows them to diffuse into peripheral target cells and is utilized for the intracellular generation of calcitriol [5,13]. This intracellular calcitriol is essential for maintaining immune cell activities, including preventing autoimmunity and combating invading pathogens [5,107,108,109]. Once generated adequately within the immune cells, calcitriol activates the cytosol’s vitamin D (calcitriol) receptors (VDR/CTR) and provides autocrine and paracrine signaling as well as genomic modulations [5].

Calcitriol concentrations in the circulation are controlled by parathyroid hormone (PTH) and ionized calcium in the blood but not by tissue 24-hydroxylase. In contrast, in the target tissues, the production of calcitriol is mainly regulated by a combination of the serum 25(OH)D concentration (and D_3_) in a concentration-dependent manner (for diffusion). It is subjected to the feedback catabolic activity of tissue 24-hydroxylases and not by PTH. Except for membrane effects, vitamin D has minimal actions; therefore, the “physiological concentrations of vitamin D status” level refers to D_3_ and 25(OH)D concentrations in the circulation, of which only the latter is measured routinely to assess vitamin D status. Peripheral target cells, like immune cells, primarily depend on the diffusion of vitamin D and 25(OH)D from the circulation to generate higher concentrations of non-hormonal calcitriol, intracellularly [13,51].

### 2.1. Mechanisms of Adequate Vitamin D Supplementation in Infections

The crucial role of vitamin D adequacy in combating acute infections was confirmed a decade ago [47,110,111]. Serum 25(OH)D concentration thresholds needed for robust immune systems to overcome infections [112,113,114] and to reduce health risks were clarified in recent years in adults [13,115,116,117] and children [113,118,119,120]. Studies have affirmed the mechanisms of action of how ultraviolet-B (UVB) rays and vitamin D supplements help individuals with infections recover faster [121,122]. Subsequently, many studies have consolidated these findings [47,123,124,125].

The converging data indicate that the minimum effective serum 25(OH)D concentration to reduce infections and their severity is 40 ng/mL (100 nmol/L) [4,108,117,126], with the optimum being above 50 ng/mL [13,47,51,111,125,127,128]. Regarding SARS-CoV-2 infection, numerous studies reported that effective use of vitamin D_3_ and calcifediol significantly improves clinical outcomes from those with hypovitaminosis D, including reduced hospitalizations and deaths [129,130,131,132,133,134,135]. The primary mechanism is stimulating the immune system, supported by other mechanisms discussed previously, including subduing the RAS and increasing the expression of ACE-2 [66].

Meta-analyses encompassing a variety of heterogeneous studies concluded that vitamin D reduces the incidence of acute respiratory illnesses [45,114,136,137] and significantly reduces the severity and mortality from COVID-19 [45,46,125,130,138,139,140,141,142]. In total, ten clinical trials out of 321 peer-reviewed clinical studies that used vitamin D as the primary intervention to investigate the effects on clinical outcomes in COVID-19 (from May 2020 to June 2024) reported a significant reduction in hospitalizations, ICU admissions, or deaths [117,131,132,135,143,144,145,146,147,148,149,150,151,152,153,154,155,156] [early therapies for COVID-19 (https://c19early.org, accessed on 25 January 2024) and publications specifically related to vitamin D (https://c19early.org/d, accessed on 25 January 2024) [48].

### 2.2. Mechanisms Lowering the Severity of Infections

Vitamin D supplementation has been shown to reduce the severity and complications of COVID-19 [157]. Hundreds of peer-reviewed publications have confirmed that serum 25(OH)D concentrations below 12 ng/mL (indicating severe vitamin D deficiency) pose a significant risk for vulnerability to SARS-CoV-2 infection [158,159,160,161], its complications, and mortality [135,162,163,164]. Supplementation with cholecalciferol (vitamin D_3_) or calcifediol [25(OH)D] rapidly elevates serum 25(OH)D concentrations and decreases the risk of complications and deaths from SARS-CoV-2 infection [52,165,166,167,168,169,170,171].

Moreover, adequate vitamin D supplementation in COVID-19 patients with co-morbidities has been observed to reduce complications, length of hospital stays, and disease severity, leading to lower mortality rates [47,150,151,152,155,172,173,174,175,176]. Given the evidence, vitamin D should be considered a crucial component of the physician’s arsenal in the fight against COVID-19 [135].

### 2.3. Vitamin D Is Essential for Activating Immune Cells

Calcitriol is the most active vitamin D metabolite, crucial for combating invading pathogens and preventing autoimmunity and chronic diseases [29,107,108]. Through multiple mechanisms, calcitriol modulates the immune system [12]. When secreted from renal tubular cells into the bloodstream, calcitriol functions as a hormone [5]. Circulatory calcitriol alters the behavior of cells involved in calcium–phosphate–bone metabolism and intestinal, bone, and parathyroid cells.

The average circulatory concentration of calcitriol in the circulation is about 0.045 ng/mL, while the concentration of its free, diffusible form is far below the threshold needed to diffuse into immune cells and initiate intracellular signaling [4,29]. Moreover, vitamin D and calcifediol [25(OH)D] concentrations in the circulation are about 900-fold higher than circulating calcitriol (ng vs. pg/L in the blood); thus, only these two compounds serve as the substrate for intracellular calcitriol generation. Consequently, circulating calcitriol has no evident impact outside the muscular-skeletal, parathyroid, and fat cells.

However, the higher nmol-range concentrations of calcitriol generated intracellularly in response to TLR signaling provide (physiological) intracellular autocrine/intracrine signaling crucial for immune functions to overcome threats like infections [177]. Consequently, a holding mechanism increases serum levels, i.e., beyond a threat like detecting unfamiliar proteins or antigens in the circulation or local tissues [178,179]. The sporadic increases in the synthesis of calcitriol and VDR in response to TLR-4 signaling ensure the formation of sufficient calcitriol–VDR complexes to modulate transcriptions and intra-cellular autocrine signaling and genomic modulation, as and when needed [178].

The mechanisms mentioned above regulate inflammation and oxidative stresses through the abovementioned mechanisms, primarily by suppressing inflammatory cytokines and enhancing the synthesis of anti-inflammatory cytokines [177]. The immunomodulatory effects of vitamin D include the activation of immune cells such as T and B cells, macrophage and dendritic cells, and the enhanced production of several antimicrobial peptides and neutralizing antibodies [108,180].

## 3. Vitamin D Deficiency and Vulnerability to Infections

Vitamin D modulates the innate and adaptive immune systems [178,181]. It enhances innate immunity via complex mechanisms [182,183], including the expression of antimicrobial peptides like cathelicidin in various cells, including keratinocytes, epithelial cells, and monocytes [184]. Vitamin D also down-regulates inflammatory responses through multiple mechanisms, including dampening the RAS [183], switching Th1 cells to Th2 and Th17 to Treg cells [185,186], and increasing the expression of MAPK phosphatase-1 (MKP-1). Consequently, the latter inhibits p38 activation and reduces pro-inflammatory cytokine production in human monocytes/macrophages, as demonstrated when stimulated with lipopolysaccharides (LPS) [187].

### 3.1. Vitamin D Deficiency Increases Infection Vulnerability

Individuals with hypovitaminosis D are at a heightened risk of developing COVID-19 (and other infections) and experiencing unfavorable clinical outcomes [157,158,188,189]. In addition to vitamin D’s robust anti-inflammatory, anti-oxidant, and antimicrobial properties [148,176], intracellularly-generated calcitriol offers other beneficial effects. These include enhanced cell repair, reduced apoptosis [190], and the protection of epithelial and vascular endothelial cells [191]. In addition, it mitigates pregnancy-associated complications [179,180] and reduces all-cause mortality [192]. As a result, maintaining vitamin D sufficiency helps in rapid recovery and reduces complications and death from SARS-CoV-2 infection [157,193].

Studies reported that calcitriol inhibits the production of pro-inflammatory cytokines via T cells. Moreover, when combined with IL-2, 1,25(OH)_2_D_3_, it promotes the regulatory function of T cells, thereby contributing to immune regulation and homeostasis [194]. Numerous studies, including meta-analyses, have shown an inverse correlation between serum vitamin D levels and the severity of COVID-19 [125,129,157,158,189,195]. Therefore, it is unsurprising that vitamin D deficiency aggravates complications from COVID-19 [157,158,188,189].

Hypovitaminosis D and SARS-CoV-2 infections weaken epithelial cell gap junctions, potentially facilitating the passage of substances, fluids, and microbes across membranes and allowing viral dissemination [196]; when combined with other micronutrients like selenium and zinc, vitamin D provides a protective effect and enhances the physiological functions of tight cell gap junctions in epithelial cells [197]. This synergy improves the effectiveness of the tissue barrier, particularly in preventing the entry and propagation of microbes, particularly viruses [198,199].

### 3.2. Hypovitaminosis D Causes Immune Cell Dysfunction

Vitamin D deficiency leads to immune dysfunction, heightening susceptibility to infections and autoimmune disorders [4]. Low vitamin D reduces ACE-2 expression and increases viral loads, replication, and dissemination [200]. It also increases the potential of mutations (also after COVID-19 vaccines) [201,202,203] and could promote the naturally evolving gain of functions in viruses. For instance, dominant mutations enhance the affinity between the spike protein receptor-binding domain (RBD) and the ACE-2 receptor molecules [203], as seen in variants like Delta and Omicron, resulting in heightened infectivity (R_0_) [204,205], as illustrated in recent dominant mutants such as Omicron BA.2 [204].

Natural and vaccine-mediated neutralizing antibodies bind either to or near the ACE-2 binding region of the RBD [206]. Consequently, critical mutations in the viral RBD interfere with their recognition by neutralizing antibodies, leading to immune evasion. Moreover, Omicron variants BA.4 and BA.5 and other newly described dominant mutations notably occur in the L452R and F486V regions of the RBD, impairing the immune neutralization of viruses [197]. These mechanisms have been documented using sera from individuals who have received triple vaccination [205].

### 3.3. Vitamin D Insufficiency and Chronic Diseases

Epidemiological and case-control studies reported strong associations between vitamin D deficiency and several immune-related chronic diseases, such as type-1 diabetes [207,208], type-2 diabetes [209], connective tissue disorders [210], inflammatory bowel disorders [211], chronic hepatitis [212], asthma [213], respiratory infections [45], and cancer [214,215]. However, interventional studies have yielded weak data primarily due to poor study designs [5,216,217,218]. With a dysfunctional immune system and low ACE-2 expression, this is unsurprising.

Aging and chronic diseases [219,220] have a higher prevalence of hypovitaminosis D, and most of these conditions have reduced ACE-2 expression [29,221]. Since ACE-2 has protective functions, it is predictable that its lower expression in aging and co-morbidities increases SARS-CoV-2-associated complications and deaths [220]. Additionally, the lower expression of ACE-2 is reported in chronic conditions such as hypertension, obesity, and diabetes [197], which are also associated with an increased risk of complications from SARS-CoV-2 infection [157,158,188,189,222,223].

Therefore, the fundamental vulnerability in aging and co-morbidities may lie in hypovitaminosis and the low expression of ACE-2, illustrating the interlink between the mentioned chronic conditions and increased infection vulnerability. These data support that the synergistic adverse effects of hypovitaminosis D and reduced ACE-2 expression in patients with chronic diseases heighten risks for symptomatic COVID-19, complications, and deaths [220,221].

### 3.4. Co-Morbidities and Disease Vulnerability

Over the last 20 years, over 8000 clinical research articles have been published illustrating a robust inverse association between serum 25(OH)D concentrations and disease vulnerability, severity, and deaths from various diseases [224,225], especially from infections [4,226,227]. Despite the country’s location, gross national products, healthcare expenditure, or access to healthcare, older people with co-morbidities [18] and institutionalized people have shown to have the highest prevalence of severe vitamin D deficiency (e.g., serum 25(OH)D concentrations less than 12 ng/mL) [162,228,229].

Co-morbidities are strongly associated with low serum 25(OH)D and ACE-2 levels [230]. These groups include nursing home residents, developmental disability centers, group homes, incarcerated people, and routine night shift workers [29]. They have the highest rates of hypovitaminosis D and experience complications [231], ICU admissions, and deaths from SARS-CoV-2—this was observed vividly during the early part of the COVID-19 pandemic [125,232,233,234,235,236,237]. One common factor in the groups mentioned above is the low expression of ACE-2 [220,231]. Therefore, supplementing them with sufficient vitamin D daily (or once a week) dampens the RAS system (thus subduing inflammation) and enhances the ACE-2 expression, reducing disease severity and co-morbidity [220,231].

Such an approach has significantly reduced symptomatic SARS-CoV-2 infections, complications, and deaths [13,234,236,238,239]. However, this expectation failed to materialize during the pandemic because health agencies neglected this important natural defense mechanism [240]. This failure to boost the immune systems with simple remedies like vitamin D, the denial of natural immunity, disallowing the use of early repurposed therapies, and reliance on vaccination alone to control the pandemic [241,242] have increased hospitalizations and deaths from COVID-19 [193,231,243].

## 4. Effects of SARS-CoV-2 on the Immune System

Having hypovitaminosis D pre-pandemic or pre-infection increases the risks and vulnerability of SARS-CoV-2 [130]. Moreover, vitamin D deficiency at the time of diagnosis of SARS-CoV-2 infection significantly increased disease severity and mortality [13,43,55,56]. In contrast, vitamin D sufficiency is protective against severe COVID-19 disease and death [13,46,57,58]. These data support the satisfaction of Bradford Hill’s criteria for establishing that hypovitaminosis D increases the risks of complications from SARS-CoV-2 [18,59,244,245]. Evidence also supports that vitamin D deficiency is a cause of vulnerability, severity, and mortality from the SARS-CoV-2 virus [13,43,46,47,48,49,50].

The RBD of the SARS-CoV-2 spike protein interacts with membrane-bound ACE-2 receptors [246,247,248]. Subsequently, the serine protease TMPRSS2 cleaves the spike protein into two functional subunits: S1 and S2. Following this cleavage, the S2 subunit undergoes a conformational change, enabling fusion with the host cell membrane and facilitating endocytotic internalization [103,249,250]. The Two-Pore Segment Channel 2 (TPC2) molecules in the endo-lysosomal system also facilitate the entry of SARS-CoV-2 into human cells [249]. This process is activated by ionized calcium (Ca^2+^), subsequently activating nicotinic acid adenine-dinucleotide phosphate (NAADP) intracellular messengers [251].

### 4.1. Physiology and Pathological Pathways of the RAS Axis

In the RAAS system, the enzyme renin (the rate-limiting step of RAS) activates angiotensinogen into angiotensin-I, and ACE-1 generates Ang-II (Figure 2). Renin catalyzes the formation of pro-peptide Ang-I and promotes the expression of ACE-1. Increased Ang-1 (relatively inactive molecules) leads to the increased synthesis of Ang-II via ACE-1—a potent vasoconstrictor peptide (Figure 2). Ang-II directly elevates peripheral vascular resistance and hypertension (especially pulmonary). Moreover, it also activates pro-coagulatory pathways, inflammatory cytokines, and interstitial fibrosis via the Ang-II receptor type 1 (AT1) receptors [93,100]. Figure 2 illustrates the normal, physiological, and alternative pathological pathways of the RAS.

The risk of the dysregulation of the RAS is high in hypovitaminosis D, which has less control over the enzyme renin. Severe coronaviral infections can lead to a cytokine storm, causing lung injury with ARDS and initiating coagulatory abnormalities [77]. SARS-CoV-2 primarily affects lung tissues, pneumocytes, alveolar interstitium, and capillaries. The alveolar epithelial cells, the port of entry of SARS-CoV-2, have high concentrations of ACE-2 receptors on their membranes [252]. ACE-2 expression is down-regulated with the viral infection, and the ACE-2/Ang(1–7)/Mas receptor (MasR) axis is suppressed [77], which augments the classic RAS, leading to diffuse inflammations and associated adverse effects. This cascade of events can cause severe inflammation, oxidative stress, and lung damage, leading to fluid extravasation into soft tissues, causing pulmonary edema and fibrosis [93,252].

### 4.2. Renin-Angiotensin System Related to SARS-CoV-2

As mentioned above, calcitriol is a crucial regulator of the RAS axis, mainly by suppressing renin gene expression, a rate-limiting step of synthesis of Ang-II via cyclic AMP (cAMP)-dependent PKA signaling [74]. Calcitriol independently increases the expression of ACE-2 [253], reducing Ang-II and increasing the generation of Ang_(1–7),_ suppressing the formation of the CRE-CREB-CBP complex [70] and keeping RAS activity under control [197]. Meanwhile, certain viral infections like SARS-CoV-2 stimulate RAS activity, aggravating the situation in those with vitamin D deficiency and making them more vulnerable to developing cytokine storms [19,70].

SARS-CoV-2 infection in individuals with severe vitamin D deficiency markedly increases the vulnerability to severe complications, including ARDS and death [254,255,256]. Low ACE-2 concentrations in such persons [256] significantly increase the susceptibility to contract viruses, especially coronaviruses, and develop complications [117,146]. Once infected, coronaviruses indirectly increase the renin and, reduce ACE-2 and soluble ACE-2 concentrations. The latter is partly caused by the consumption (i.e., internalization and destruction) of membrane-bound ACE-2 receptors [59]. Low ACE-2, in conjunction with the over-activation of the RAS [55], leads to the excess and uncontrolled production of Ang-II, leading to severe adverse effects, increasing complications, the risk for cytokine storms, and death [19,197].

Also, widespread inflammation and oxidative stress injure the pulmonary epithelial and vascular endothelial cells and their basement membranes [257], impairing tight junction, with virus dissemination and fluid leakage into soft tissues [257], including the lungs, pericardium, intestine, and brain [29,196,258]. Pulmonary epithelial damage causes hypoxia and increases the risk of pneumonia and ARDS [69,70]. The endothelial abnormalities lead to micro-thrombosis, embolization, and intravascular thrombosis [259,260]. Figure 3 illustrates the RAS axis in a normal physiological state, in an activated state with hypovitaminosis D, and in the presence of SARS-CoV-2 infection [19].

### 4.3. Regulation of Inflammation by Vitamin D via the RAS

Vitamin D is a potent negative endocrine regulator in the RAS through a canonical pathway [197]. It achieves this by inhibiting the cAMP response element-binding protein (CREB), a key transcription factor for renin gene regulation [74]. Vitamin D-driven suppression of renin [261] reduces RAS activity, such as lowering the molecular expression of ACE-1 and Ang-II while increasing the expression of ACE-2 [74,253]. These effects have been observed both in vitro and in vivo [253]. Notably, low serum levels of 25(OH)D concentrations are inversely correlated with higher RAS activity, elevated plasma renin activity, and Ang-II levels, contributing to increased blood pressure [262,263,264].

In studies using LPS-exposed rat lung tissue, calcitriol increased ACE-2 expression while reducing renin, angiotensin II, ACE, and AT1 receptor expression. Additionally, calcitriol exhibited a dose-dependent reduction in the permeability of blood vessels in rat lungs, mitigating damage induced by LPS [253]. Activating the ACE-2/Ang_(1–7)_/MAS axis by vitamin D further enhances the production of ACE-2 and Ang_(1–7)_ concentrations. Ang_(1–7)_’s vasodilatory, anti-inflammatory, and anti-thrombotic effects help counteract hypertension, inflammation, and coagulatory abnormalities [265,266,267] (Figure 2), mitigating the adverse effects caused by SARS-CoV-2 and its spike protein.

As previously mentioned, vitamin D also blocks viral entry, strengthens immunity by tightening epithelial gap junctions, and inhibits SARS-CoV-2 transcription enzymes [60], thereby combating the viral infection. Vitamin D deficiency over-activates the RAS with the inefficient counter-regulatory activation of ACE-2/angiotensin_(1–7)_/Mas axis [100,268], increasing the risks for ARDS [197]. Consequently, vitamin D deficiency is a critical component that exacerbates COVID-19 via the over-activation of RAS with an excess generation of Ang-II [197].

Vitamin D and inhibitors of ACE-1 and Ang receptor blockers work together indirectly to decrease renin synthesis, thereby reducing the activity of the RAS and the synthesis of Ang-I and Ang-II [66]. Angiotensin receptor blockers (ARBs) also inhibit the generation of Ang-II and decrease the stimulation of the angiotensin-1 (AT1) receptor. It has also been proposed that CD147/Basigin receptors bind to epitopes on the spike protein-S of the SARS-CoV virus, facilitating viral entry through endocytosis [269]. Therefore, blocking CD147/Basigin could be a target for drug development against COVID-19 [270]. Moreover, vitamin D indirectly regulates SARS-CoV-2 by modulating ACE-2 and CD147 receptor molecules [271,272]. The involvement of vitamin D and ACE-2 in modulating the SARS-CoV-2 virus [197] and its effects on the RAS are illustrated in Figure 4.

Figure 3 and Figure 4 illustrate multiple mechanisms of SARS-CoV-2 that dysregulated the RAS, reducing ACE-2 and increasing Ang-II levels. Vitamin D, ARBs, and ACEi can intervene positively in these cycles initiated by the virus, particularly in individuals with severe vitamin D deficiency. Vitamin D plays a crucial role by modulating the RAS, inhibiting Ang-II synthesis, and up-regulating ACE-2 expression, which mitigates the harmful effects of SARS-CoV-2 [19].

Ang-II type 1 receptor blockers (ARBs) and ACEi medications influence RAS, potentially mitigating the dysregulation caused by SARS-CoV-2 [19]. Together, these interventions aim to dampen the vicious cycles initiated by the virus, potentially reducing the morbidity and mortality associated with COVID-19, especially in individuals with severe vitamin D deficiency. ACEi and ARBs are indicated clinically in hypertension and other cardiovascular diseases [197]. These agents have been shown to mitigate acute lung injury by restoring the balance between two regulatory processes. Preclinical and clinical studies support the evidence of RAS disequilibrium in COVID-19 and the beneficial role of RAS modulation [252].

## 5. ACE-2 Receptor, Vitamin D, and SARS-CoV-2

Following the internalization of the ACE-2 receptor complex, the SARS-CoV-2 virus replicates using the host’s cell machinery [273]. The increased infectivity (R_0_) of SARS-CoV-2 and its mutants is attributed partly to the heightened affinity of their spike proteins to cell surface ACE-2 receptors [204,274]. Therefore, ACE-2 receptors are potential targets for drug development to prevent SARS-CoV-2 from entering human cells [249,251,274]. Respiratory tract epithelial cells co-expressing ACE-2 and TMPRSS2 molecules are the primary targets for SARS-CoV-2 entry [275]. The blockage of TMPRSS2 could serve as a molecular target for new drug development to prevent cellular access by COVID-19 [276].

### 5.1. Reduction in Viral Load through Soluble ACE-2

In the circulation, soluble ACE-2 molecules bind with SARS-CoV-2, facilitating the transport of these complexes to natural killer (NK) cells and macrophages for subsequent destruction [59]. This neutralization of viral particles prevents them from reaching membrane-bound ACE-2 receptors in lung and vascular epithelial cells. Since viruses cannot replicate outside of host cells, the binding of SARS-CoV-2 to soluble ACE-2 in the extra-cellular fluid could inhibit the replication of SARS-CoV-2 [37]. Contrary to earlier publications, the in vivo up-regulation of ACE-2 does not exacerbate but mitigates the effects of SARS-CoV-2 [277,278].

Studies have demonstrated that soluble ACE-2 can reduce viral load in vitro and protect against infection in preclinical models. For instance, Monteil et al. (2020) showed that human recombinant soluble ACE-2 (hrsACE-2) significantly blocked the initial stages of SARS-CoV-2 infections in engineered human blood vessels and kidney organoids, highlighting its potential as a therapeutic agent [104]. In contrast, some research indicated pharmacological strategies to reduce ACE-2 expression, such as using ursodeoxycholic acid (UDCA) to reduce the infection rate of SARS-CoV-2 by targeting the farnesoid X receptor (FXR) [279]. Further research is ongoing to evaluate the efficacy and safety of soluble ACE-2 in clinical settings, aiming to offer a novel approach to mitigate the impact of COVID-19 [59].

Nevertheless, it is notable that the levels of soluble ACE-2 required to inhibit SARS-CoV-2 infection may be above physiological levels. Soluble ACE-2 could enhance SARS-CoV-2 [19] infection at physiological levels by forming a complex with the virus that enters cells via endocytosis through the AT-1 surface receptor [280]. Overall evidence suggests that while increased ACE-1 activity may be harmful, the increased expression of ACE-2 is beneficial in controlling coronaviruses [281,282]. Therefore, discontinuing ACE inhibitors (ACEi) and angiotensin receptor blockers (ARBs) solely due to COVID-19 infection is not recommended [19].

### 5.2. Reduction Consequences of the Lower Expression of ACE-2

SARS-CoV-2 also reduces ACE-2 expression, further hampering protective physiological functions of the ACE-2/angiotensin_(1–7)_/Mas axis [100,268] and increasing the risk for pulmonary and cardiac complications [278]. Despite using the ACE-2 receptor by SARS-CoV-2 S-protein to enter human cells, up-regulation does not increase infection vulnerability [19,278]. Meanwhile, hypovitaminosis D not only reduces ACE-2 synthesis but also exacerbates the over-production of Ang-II [19,283]. These increase the risks for abnormal coagulation and interstitial fibrosis [93]. The increased Ang-II creates pathologic vasoconstriction, causing pulmonary hypertension (Figure 4), and liberates excessive amounts of harmful cytokines, causing systemic inflammation that could lead to cytokine storms [18].

Young children have been observed to have lower expression levels of ACE-2 in their nasal respiratory tract epithelia than adults [197]. Except for those who are immuno-suppressed or have hypovitaminosis D, children have stronger innate immunity. Consequently, they are less susceptible to severe COVID-19 complications [284]. The binding process of SARS-CoV-2 also involves cell-membrane-bound heparan sulfate proteoglycans (HSPG). Lactoferrin, a nutrient found in mammalian milk, interferes with SARS-CoV-2 binding to HSPG and membrane-bound ACE-2 receptors, potentially reducing coronaviruses’ entry into human host cells [285].

SARS-CoV-2 infection significantly reduces ACE-2 concentrations [197] and disrupts the balance of ACE-2/ACE-1 ratio. It, initiating a pathological cycle, worsens in those with hypovitaminosis D. Besides, SARS-CoV-2 up-regulates metallopeptidase ADAM17 molecules, enhancing the RAS activity and increasing the production of the inflammatogenic Ang-II [286,287,288,289,290].

Unlike epithelial and immune cells, erythrocytes and platelets lack membrane-bound ACE-2 receptors [291]. However, the interaction of the spike protein with CD147 receptors on platelets and erythrocyte membranes can lead to platelet aggregation and erythrocyte abnormalities [271,292]. SARS-CoV-2-induced damage to gap junctions and epithelial barriers results in the loss of epithelial cell integrity, which is aggravated in the presence of hypovitaminosis D. This damage also affects endothelin function, impairing gas exchange in pulmonary epithelia [100,248,276,277,278,284,285,293].

### 5.3. Restriction of Generic Medication Use and Conflicts of Interest

Despite the availability of over 200 independent RCTs and extensive data on vitamin D related to preventing and treating COVID-19 [200,294], regulatory agencies withheld approvals for the use of vitamin D [240,295]. Preventing access and use of widely available, cost-effective, generic agents, such as vitamin D and ivermectin [296,297], may have proven detrimental to patient welfare [193,242,295], increasing hospitalizations and deaths [48,132,240].

The actions mentioned above may have been driven by the desire to maintain Emergency Use Authorization (EUA) for vaccines and anti-viral agents [298]. However, this created a scenario for the approval and use of COVID-19 vaccines and anti-viral agents under (EUA), but not widely available, cost-effective generic agents like vitamin D and ivermectin [48,193,242,298]. The lack of approvals for repurposed early therapies may have harmed people [48,132,240,294,296,297,299]. 

### 5.4. ACE-2—A Double-Edged Sword in SARS-CoV-2 Infection

Whether ACEi and ARBs play a harmful or helpful role in COVID-19 remains controversial [300,301]. Theoretically, it has been suggested that increased ACE-2 membrane receptors on epithelial cells could potentially increase the cellular entry of SARS-CoV-2 [274,302]. Consequently, it has been hypothesized that the increased ACE-2 receptors resulting from ARBs, ACEi, or vitamin D up-regulation might enhance SARS-CoV-2 entry via epithelial cells [278], thereby increasing cellular infection [248]. However, subsequent data did not support the view that ACEi and ARBs increase the risk of SARS-CoV-2 infection or worsen COVID-19 outcomes [19,277,281].

Soluble ACE-2 can effectively mimic the membrane-bound ACE-2, to which the SARS-CoV-2 spike protein binds, sequestering the virus and inhibiting its ability to infect host cells. The excess synthesis of ACE-2 soluble receptor spills into the bloodstream, potentially decoying and removing the SARS-CoV-2 virus and reducing viral load. Because of the mechanisms of action, soluble ACE-2 has emerged as a potential therapeutic strategy to neutralize SARS-CoV-2 (and other coronaviruses) in the bloodstream by acting as a decoy that binds the virus and prevents it from entering human cells [19]. These findings collectively highlight the potential of soluble ACE-2 and other ACE-2-targeted therapies in mitigating the impact of COVID-19 [197].

### 5.5. Importance of Strengthening the Immune System to Overcome Infections

The body’s initial defense against invading pathogens is the innate immune system that relies on adequate vitamin D levels to function effectively [13,51]. Ensuring vitamin D sufficiency in the population is a quick and cost-effective approach to promoting a robust immune system. Additionally, optimal immune function requires other micronutrients, such as magnesium, zinc, omega-3 fatty acids, vitamin K_2_, resveratrol, and quercetin [67], and comprehensive mental and physical health support. By addressing these nutritional and lifestyle factors, individuals can enhance their immune response and overall well-being.

The mentioned approach would ensure the maintenance of a robust innate immune system by preserving sufficient vitamin D, calcitriol receptors (CTR/VDR), and *CYP27B1* expression through the production of calcitriol within immune cells [5,105]. The combined deficiencies of vitamins D and K_2_ could elevate the risk of cardiovascular events, adverse cardiac remodeling [303], and mortality and add to all-cause mortality compared to individuals with adequate levels of both vitamins. Therefore, maintaining sufficient levels of both vitamins D and K is crucial for overall health and reducing the risk of adverse outcomes [304].

Strengthening the immune system is crucial for protecting against SARS-CoV-2 infections and reducing the risk of high viral loads, which can lead to viral mutations with greater infectivity and immune evasion capabilities [193,242]. Numerous RCTs and meta-analyses have concluded that vitamin D supplementation protects against acute respiratory tract infections, particularly in individuals with profound hypovitaminosis D [305]. Maintaining adequate physiological levels of 25(OH)D supports a robust immune function that reduces the risk of respiratory infections, including those caused by SARS-CoV-2 [189].

## 6. Discussion

Numerous studies have explored the complex interactions between vitamin D and the immune system. However, it is essential to note that this article has focused on highlighting selected critically relevant studies. As previously mentioned, vitamin D is crucial in modulating and strengthening innate and adaptive immune systems. It achieves this by up-regulating or down-regulating the transcription of target genes [12] through the calcitriol receptor (CTR). Additionally, vitamin D prevents the weakening of epithelial cell barriers [190], induces the expression of antimicrobial peptides such as cathelicidin, and up-regulates MKP-1 to inhibit inflammatory cytokines [306]. Furthermore, in conjunction with interleukin-2, vitamin D promotes T cell regulation, further contributing to immune system modulation [194].

Strengthening the immune system protects against SARS-CoV-2 infections and greater viral loads, preventing the survival of viral mutations with greater infectivity and immune evasion capabilities. In addition to these general immune benefits, vitamin D and lumisterol inhibit SARS-CoV-2 transcription enzymes, reducing viral replication and infection [307]. Studies, including meta-analyses, have concluded that vitamin D supplementation protects from acute respiratory tract infection, especially in those with profound hypovitaminosis D [307].

Vitamin D counteracts the vasoconstriction, pulmonary hypertension, coagulation, and interstitial fibrosis caused by SARS-CoV-2 by inhibiting the transcription factor CREB to down-regulate renin [93], the rate-controlling step of the RAS system, resulting in reduced Ang-II production [66]. Secondly, vitamin D up-regulates ACE-2 [253], which counteracts Ang II. Converting Ang II into Ang_(1–7)_ by ACE-2 makes less Ang-II available. This reduces Ang-II levels and causes the production of Ang_(1–7)_, which, via its MAS receptor, exhibits vasodilatory, anti-inflammatory, and anti-thrombotic effects to prevent pulmonary injuries.

Vitamin D enhances the expression of ACE-2, but some have raised concerns that this could exacerbate the disease caused by SARS-CoV-2, as the virus exploits membrane-bound ACE-2 to invade epithelial cells [274,308,309,310]. However, as described above, this potential risk is mitigated by the excess synthesis of ACE-2 molecules, which spill over into the bloodstream as ACE-2-soluble decoy receptors. These soluble ACE-2 receptors bind to SARS-CoV-2 viruses and escort them to natural killer cells and macrophages for destruction [256]. This process has been confirmed in in vitro systems [104]. If this also occurs in vivo, this process will help reduce infection and viral load, minimizing complications and deaths from SARS-CoV-2 [19,286,311,312].

Therefore, rather than the associated risk of facilitating an increased entry of SARS-CoV-2 viruses, the up-regulation of ACE-2 by vitamin D sufficiency may reduce viral load, thus mitigating the effects of SARS-CoV-2 and decreasing the risks of complications from SARS-CoV-2 [259,313,314]. However, further investigation is required to gain an in-depth understanding and confirm if soluble ACE-2 is inhibitory at physiological levels [280]. These anti-viral processes are enhanced by having sufficient cofactors like magnesium, iodine, and selenium, which help maintain tight cell junctions in epithelial cells [198]. Vitamin D-mediated tight gap-junctions help to preserve cellular function and prevent excessive fluid diffusion and viral spread across cell membranes [59].

ACE-inhibiting enzymes and ARBs can prove beneficial by reducing Ang-II production and AT1 receptor activation and up-regulating soluble ACE-2 [277,281]. However, we note limited studies on the direct relationship between vitamin D and SARS-CoV-2 replication. In conclusion, overall data support that vitamin D sufficiency, ACE inhibitors, and ARBs reduce the risk of COVID-19, associated complications, and deaths [19,259,286,313]. 

## 7. Conclusions

The authors recommend ensuring the vitamin D sufficiency of patients with supplementation to rapidly raise and maintain serum 25(OH)D concentrations above 50 ng/mL for a robust immune system to protect against coronavirus [51], especially COVID-19, and other viral diseases (e.g., dengue) to mitigate morbidity/complications and mortality. Vitamin D is required for effective innate and adaptive immune system function and counteracts the pathological effects of over-stimulated RAS by SARS-CoV-2. Its beneficial actions include lowering renin and up-regulating ACE-2, which lowers Ang II and increases vasodilator Ang_(1–7)_. The latter also has anti-inflammatory and anti-coagulant properties to mitigate the harmful effects of SARS-CoV-2, especially in the lungs and vascular system—vitamin D’s upregulation of soluble ACE-2 assists in the elimination of SARS-CoV-2, thus reducing viremia. ACEi and ARBs appear to contribute to partly relieving the adverse effects of SARS-CoV-2.

## Figures and Tables

**Figure 1 biology-13-00831-f001:**
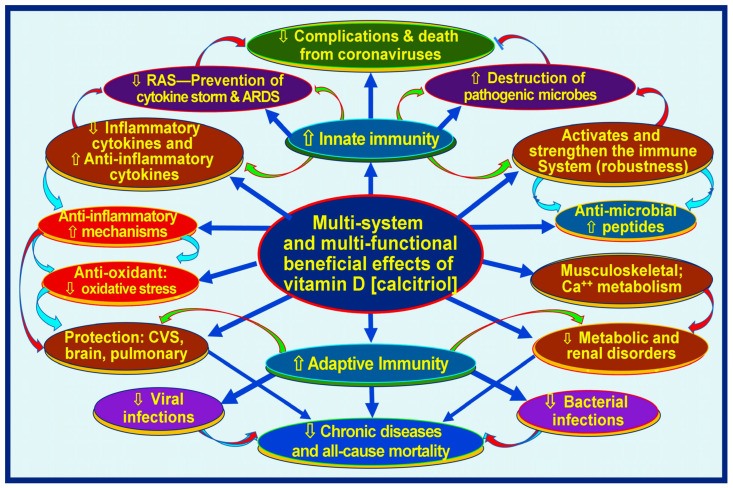
Infections and immune-related broader functions of vitamin D (calcitriol, 1,25(OH)_2_D). The figure illustrates muti-system-wide functions of vitamin D related through the modulation of innate and adaptive immune systems, resulting in lowering complications from infections and chronic disease burdens [⇧ = increased; ⇩ = reduced; RAS: renin-angiotensin-system; CVS: cardiovascular system] (after Wimalawansa, Nutrients, 2022) [51].

**Figure 2 biology-13-00831-f002:**
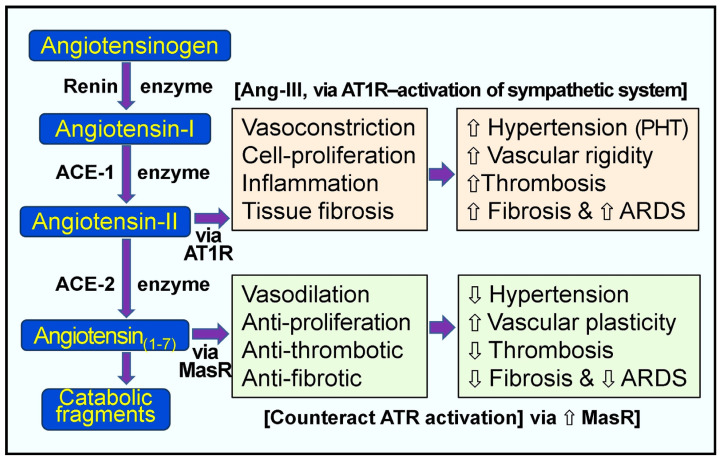
Pathological and physiological responses of the renin-angiotensin system. Peach and green boxes illustrate the renin-angiotensin system’s regulatory and counter-regulatory physiologic pathways. When excess angiotensin-II (Ang-II) is synthesized, as in the case of hypovitaminosis D and SARS-CoV-2 infection, this leads to the over-activation of the AT1 receptors (AT1-R) with pathological manifestations, as indicated in the peach colored boxes [⇧ = increased; ⇩ = reduced; ARDS = acute respiratory distress syndrome; RAS, renin-angiotensin system; ACE, angiotensin-converting enzyme; ACE-2, angiotensin-converting enzyme 2; Ang 1–7, angiotensin 1–7; Ang-I, angiotensin-I; Ang-II, angiotensin-II; AT1R, type 1 angiotensin-II receptor; MasR, MAS proto-oncogene receptor. PHT, pulmonary hypertension].

**Figure 3 biology-13-00831-f003:**
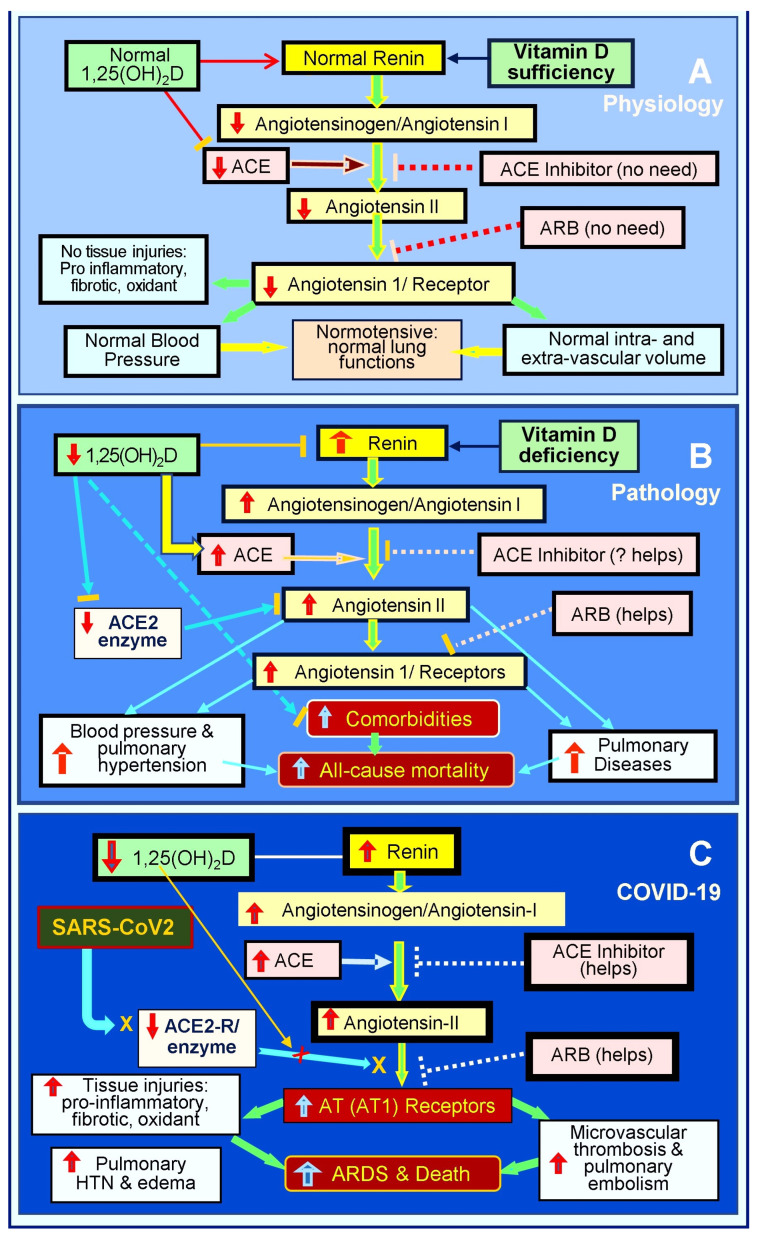
This diagram outlines the status of the renin-angiotensin axis (RAS) axis: (**A**) physiological status, (**B**) pathological/activated status in the presence of vitamin D deficiency, and (**C**) following SARS-CoV-2 infection. RAS axis homeostasis is disrupted by hypovitaminosis D. SARS-CoV-2 or other coronal viral infections markedly activate the RAS, leading to pathologically elevated levels of angiotensin -II and the suppression of ACE-2. This hyperactivation of the RAS leads to increased complications and mortality (⇧ = increased; ⇩ = reduced; ACE: angiotensin-converting enzyme; ARB: angiotensin receptor blockers; AT1R: type 1 angiotensin-II receptor; ARDS: acute respiratory distress syndrome).

**Figure 4 biology-13-00831-f004:**
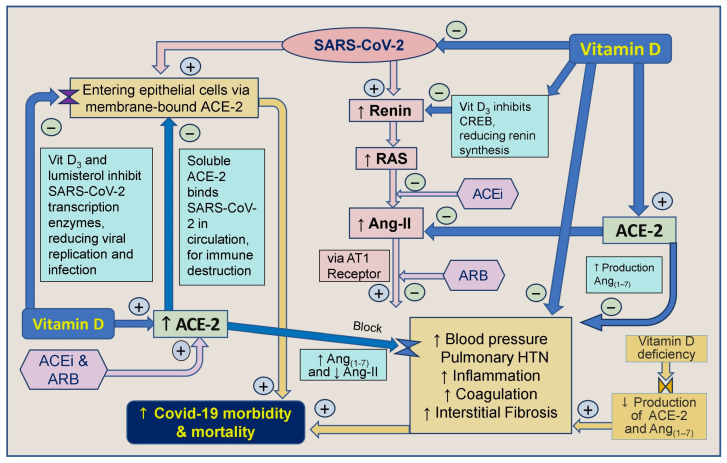
Vit D strengthens innate and adaptive immune systems. This summary outlines the correlation between vitamin D, angiotensin-converting enzyme-2 (ACE-2), angiotensin-converting enzyme inhibitors (ACEi), and angiotensin II receptor blockers (ARBs) concerning severe acute respiratory syndrome coronavirus-2 (SARS-CoV-2) and their impact on COVID-19 morbidity and mortality ([↑ = increased; ↓ = reduced; RAS: renin-angiotensin-system; CVS: cardiovascular system; ACE: angiotensin-converting enzyme; ARB: angiotensin receptor blockers; AT1R: type 1 angiotensin-II receptor; ARDS: acute respiratory distress syndrome; HTN: hypertension).

**Table 1 biology-13-00831-t001:** Highlighting the critical functions of the RAS ^#^.

**Physiological Function**	**Brief Description of Role, Regulations, and Actions**
Blood pressure	The RAS regulates blood pressure by controlling the vascular tone via the constriction of blood vessels. Angiotensin II is a potent vasoconstrictor peptide—a product in the RAS system that increases blood pressure.
Fluid and electrolyte balance	The RAS influences and helps maintain the body’s sodium and water balance. Angiotensin II stimulates the release of aldosterone from the adrenal cortex, which promotes sodium reabsorption and potassium excretion in the kidneys. This sodium retention leads to secondary water retention, increasing blood volume and blood pressure.
Blood volume	By regulating sodium and water reabsorption in the kidneys, the RAS helps maintain the overall blood volume. This is critical for maintaining adequate perfusion pressure and ensuring sufficient blood flow to vital organs.
Systemic vascular resistance	The constriction of systemic arterioles by angiotensin II increases peripheral resistance, which is a major determinant of blood pressure.
Renal function	The RAS modulates glomerular filtration rate (GFR) and renal blood flow. Angiotensin II constricts efferent arterioles in the kidneys, helping maintain GFR despite systemic blood pressure changes.
Cardiac function	RAS affects cardiac function by influencing myocardial contractility and promoting cardiac hypertrophy. Chronic activation of the system can lead to pathological changes in the heart, such as ventricular hypertrophy and fibrosis, contributing to developing heart failure.
Immune regulation	RAS plays a critical role in immune homeostasis. Vitamin D modulates this activity. The over-activity of the RAS could cause the excess generation of inflammatory cytokines, excess angiotensin-II, and generalized inflammation. When uninhibited, it can lead to a cytokine storm.

^#^ The above summary was created based on physiological and pathological findings and understanding from multiple publications.

## Data Availability

Data are contained within the article.

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
