# Peer review of "Unveiling the Interplay—Vitamin D and ACE-2 Molecular Interactions in Mitigating Complications and Deaths from SARS-CoV-2"

_biology, 2024, doi:10.3390/biology13100831_

Round 1
Reviewer 1 Report
Comments and Suggestions for Authors
I thoroughly read the review "Unveiling the Interplay—Vitamin D and ACE-2 Molecular Interactions in Reducing 2 Complications and Deaths from SARS-CoV-2" and found it interesting and has public health importance. Overall, I suggest improvement in writing.
Specific comments:
Abstract
-Start with a general sentence like the importance of Vitamin D and ACE-2 and link with SARS-CoV2.
-One-sentence objectives and methods (search strategies of articles)
-Add a concluding sentence at the end of the abstract.
Highlights
-If this section is mandatory for this journal, keep it; otherwise, delete it.
Introduction
Line 50-56: Use references for those statements.
Line 83: Use the hypothesis/aims of this study.
Methods
-Add methodology by defining how you use the search engine of the articles, how many authors are involved in searching articles ( to avoid bias), and which referencing tools are used.
Figures 1 and 3: avoid dark colours and use light colours for these figures.
Line 88: Rewrite this sentence.
Table 1: Use the source of Table 1 as I see this table is directly/partly from others.
Line 299: Avoid the link and use reference only.
Points 3.3 and 3.4: delete these 2 sections.
Glossary/Abbreviations: Delete this section if it is not mandatory.
Comments on the Quality of English Language
English Check required.
Author Response
Reviewers comments and Author responses:
Reviewers’ comments are indented, and author responses are in bold format for easier identification.
Comments and Suggestions for Authors
Reviewer 1:
I thoroughly read the review “Unveiling the Interplay—Vitamin D and ACE-2 Molecular Interactions in Reducing 2 Complications and Deaths from SARS-CoV-2” and found it interesting and has public health importance. Overall, I suggest improvement in writing.
Specific comments:
Abstract
-Start with a general sentence like the importance of Vitamin D and ACE-2 and link with SARS-CoV2.
Thank you. As suggested, the authors have introduced a new sentence at the beginning of the abstract.
-One-sentence objectives and methods (search strategies of articles)
-Add a concluding sentence at the end of the abstract.
The authors also have modified the abstracts to encompass the above.
Highlights
-If this section is mandatory for this journal, keep it; otherwise, delete it.
Highlights are vital to these types of articles, which benefit readers who are less familiar with cutting-edge science.
Introduction
Line 50-56: Use references for those statements.
Thank you; the authors added new references to strengthen this paragraph.
Line 83: Use the hypothesis/aims of this study.
The authors added a couple of new sentences to highlight the hypothesis/aim of this study. The last paragraph of the introduction section.
Methods
-Add methodology by defining how you use the search engine of the articles, how many authors are involved in searching articles ( to avoid bias), and which referencing tools are used.
The authors respectfully state that this is not a Systematic Review. Thus, including such a detailed investigating methodology (although we adhered to basic PRISMA guidelines and searched three research databases) is inappropriate for this review.
Figures 1 and 3: avoid dark colours and use light colours for these figures.
As suggested, the background colours were changed to light colours in those figures.
Line 88: Rewrite this sentence.
The authors have modified the said sentence.
Table 1: Use the source of Table 1 as I see this table is directly/partly from others.
This is a brand new table created by the authors, using a basic physiological understanding of the functions of the RAS. Therefore, no need to provide any reference.
Line 299: Avoid the link and use reference only.
The authors believe that this particular weblink is vital for the clarity of the paragraph.
Points 3.3 and 3.4: delete these 2 sections.
The authors believe the two sections mentioned above are crucial for readers to understand the relationships between “vitamin D deficiency,” reduced ACE-2 levels, and increased vulnerability to infections and autoimmune disorders. Conversely, vitamin D sufficiency helps minimize these ailments.
Similarly, vitamin D deficiency is associated with an increased prevalence and severity of chronic diseases and co-morbidities. Both conditions are known to elevate RAS activity and reduce ACE-2 levels. Therefore, these sections are essential for comprehending the concepts and outcomes of infections like SARS-CoV-2.
Nevertheless, we have modified both sections to focus on vitamin D and ACE-2 and reduced the word count.
Glossary/Abbreviations: Delete this section if it is not mandatory.
The authors believe that the abbreviations are helpful for readers who may not be familiar with the concepts discussed in this paper. Including them allows readers to refer to the sections at the end, facilitating a better understanding of the review article's content.
Thank you very much for the input and suggestions to improve the quality of the manuscript.
Greatly appreciated.
Reviewer 2 Report
Comments and Suggestions for Authors
In the present article, the authors describe different ACE-Vitamin interaction pathways with SAR-CoV2. Overall the review article is well written and will be interesting for scientific readers.
Minor Comments:
· The authors should highlights the future prospects of ACE-Vitamin interaction with SAR-CoV2 which are still to be explored.
· The colour scheme of the all figures may be modified i-e light background can be used to make it more understandable and attractive to readers.
· Data in the figures are adopted from various articles, so it is suggested to give their references (i-e adopted from..) in figure legends.
· Abbreviations or symbols (for increase and decrease) used in the figures should be defined in the figure 1 legend.
· Delete the heading Renin-angiotensin system given inside the figure 2.
Author Response
Reviewers comments and Author responses:
Reviewers’ comments are indented, and author responses are in bold format for easier identification.
Comments and Suggestions for Authors
Reviewer 2
Comments and Suggestions for Authors
In the present article, the authors describe different ACE-Vitamin interaction pathways with SAR-CoV2. Overall the review article is well written and will be interesting for scientific readers.
Minor Comments:
- The authors should highlight the future prospects of ACE-Vitamin interaction with SAR-CoV2 which are still to be explored.
- The colour scheme of the all figures may be modified i-e light background can be used to make it more understandable and attractive to readers.
Thank you—We have done that by changing/ making them have lighter backgrounds to highlight the figure contents.
- Data in the figures are adopted from various articles, so it is suggested to give their references (i-e adopted from..) in figure legends.
Figures except Figure 1 (reproduced from Nutrients, 2022, we have already acknowledged them with the reference); others are original and thus need no references.
- Abbreviations or symbols (for increase and decrease) used in the figures should be defined in the figure 1 legend.
Yes, indeed: we have now corrected this, as the referee rightly stated. Thank you.
- Delete the heading Renin-angiotensin system given inside the figure 2.
Reasonable suggestion—Thank you:
Readers who are unfamiliar with the RAS system, how it interacts with the other components, and their physiological and pathological contributions need a better description. As requested, this title was deleted and added to the Figure legend for clarity.
Thank you very much for the input and suggestions to improve the quality of the manuscript.
Greatly appreciated.
Round 2
Reviewer 1 Report
Comments and Suggestions for Authors
-The manuscript was revised well. Some spelling and English checks are still required before publication.
Comments on the Quality of English LanguageMinor English check required.
Author Response
The manuscript was revised well. Some spelling and English checks are still required before publication.
Author's reply: Thanks. I have modified the Highlights, and a few errors and typos were detected in the manuscript.